# Immunomodulatory Effects of Stereotactic Body Radiotherapy and Vaccination with Heat-Killed Mycobacterium Obuense (IMM-101) in Patients with Locally Advanced Pancreatic Cancer

**DOI:** 10.3390/cancers14215299

**Published:** 2022-10-27

**Authors:** Freek R. van ‘t Land, Sai P. Lau, Willem de Koning, Larissa Klaase, Madelief Vink, Anneloes van Krimpen, Jasper Dumas, Disha Vadgama, Joost J. Nuyttens, Dana A. M. Mustafa, Ralph Stadhouders, Marcella Willemsen, Andrew P. Stubbs, Joachim G. Aerts, Casper H. J. van Eijck

**Affiliations:** 1Department of Surgery, Erasmus University Medical Center, ‘s-Gravendijkwal 230, 3015 CE Rotterdam, The Netherlands; 2Department of Pathology and Clinical Bioinformatics, Erasmus University Medical Center, ‘s-Gravendijkwal 230, 3015 CE Rotterdam, The Netherlands; 3Department of Pulmonary Medicine, Erasmus University Medical Center, ‘s-Gravendijkwal 230, 3015 CE Rotterdam, The Netherlands; 4Department of Pathology, the Tumor Immuno-Pathology Laboratory, Erasmus University Medical Center, ‘s-Gravendijkwal 230, 3015 CE Rotterdam, The Netherlands; 5Department of Radiation Oncology, Erasmus University Medical Center, ‘s-Gravendijkwal 230, 3015 CE Rotterdam, The Netherlands; 6Department of Cell Biology, Erasmus University Medical Center, ‘s-Gravendijkwal 230, 3015 CE Rotterdam, The Netherlands; 7Erasmus MC Cancer Institute, Erasmus University Medical Center, ‘s-Gravendijkwal 230, 3015 CE Rotterdam, The Netherlands

**Keywords:** pancreatic ductal adenocarcinoma (PDAC), locally advanced pancreatic cancer (LAPC), stereotactic body radiotherapy (SBRT), mycobacterium vaccines, cancer immunotherapy, immuno-oncology

## Abstract

**Simple Summary:**

Around thirty-five percent of pancreatic cancer patients present with locally advanced pancreatic cancer. These patients are treated with chemotherapy, and sometimes (stereotactic)radiotherapy can be added to the treatment regimen. In this study, we treated patients with locally advanced pancreatic cancer, after standard-of-care treatment with chemotherapy, with stereotactic body radiotherapy and an immunological adjuvant called IMM-101. We hypothesized that this combination treatment has the potential to induce a potent anti-tumor immune response. This study aimed to investigate the safety and immuno-modulatory effects of the treatment in the peripheral blood. The treatment demonstrated to be safe. Immune monitoring of the peripheral blood showed transient lymphodepletion and signs of immune activation after treatment. Moreover, immune activation after treatment correlated with improved progression-free survival.

**Abstract:**

Background: Patients with locally advanced pancreatic cancer (LAPC) are treated with chemotherapy. In selected cases, stereotactic body radiotherapy (SBRT) can be added to the regimen. We hypothesized that adding an adjuvant containing a heat-killed mycobacterium (IMM-101) to SBRT may lead to beneficial immuno-modulatory effects, thereby improving survival. This study aims to investigate the safety of adding IMM-101 to SBRT and to investigate the immuno-modulatory effects of the combination treatment in the peripheral blood of LAPC patients. Methods: LAPC patients were treated with SBRT (40 Gy) and six intradermal vaccinations of one milligram IMM-101. The primary endpoint was an observed toxicity rate of grade 4 or higher. Targeted gene-expression profiling and multicolor flow cytometry were performed for longitudinal immune-monitoring of the peripheral blood. Results: Twenty patients received study treatment. No treatment-related adverse events of grade 4 or higher occurred. SBRT/IMM-101 treatment induced a transient decrease in different lymphocyte subsets and an increase in CD14+CD16−CD11b+HLA−DR^low^ myeloid-derived suppressor cells. Importantly, treatment significantly increased activated ICOS+, HLA-DR+ and Ki67+PD1+ T and NK cell frequencies. This was not accompanied by increased levels of most inhibitory markers, such as TIM-3 and LAG-3. Conclusions: Combination therapy with SBRT and a heat-killed mycobacterium vaccine was safe and had an immune-stimulatory effect.

## 1. Introduction

Pancreatic ductal adenocarcinoma (PDAC) is a notoriously lethal malignancy with a five-year survival rate of less than 5% [1]. About thirty-five percent of patients present with locally advanced pancreatic cancer (LAPC) [2]. LAPC is treated with induction chemotherapy, preferably with the multi-agent FOLFIRINOX regimen in young and fit patients [3]. Next to FOLFIRINOX, gemcitabine combined with nab-paclitaxel is another adequate first-line treatment option, which is often better tolerated than FOLFIRINOX [3]. Stereotactic body radiotherapy (SBRT) can be added to the treatment regimen if there are no signs of disease progression after the chemotherapy [4,5,6].

Radiation therapy is the cornerstone of treatment for many cancer types, with fifty percent of cancer patients being treated with some form of radiotherapy throughout their illness [7]. Traditionally, radiation therapy has been utilized for its direct cytotoxic properties, inducing tumor cell apoptosis [8]. However, besides the direct cytotoxic effect, there is emerging evidence that radiation, particularly SBRT, has potential immuno-modulatory effects. Upregulation of immunogenic cell surface markers such as ICAM-1, MHC-1 and Fas on tumor cells has been described following radiotherapy [9,10,11,12,13]. Cancer cells may escape immune-surveillance trough the downregulation of MHC-1 molecules [14]. The upregulation of MHC-1 molecules by radiation therapy may revert this escape mechanism. Additionally, irradiation can induce an upregulation of FAS molecules on tumor cells, thereby improving the cytotoxic efficacy of T cells [12]. Moreover, radiotherapy has been demonstrated to be able to induce immunogenic cell death [15], thereby reinforcing the cancer-immunity cycle [16,17]. In our previous LAPC-1 trial, LAPC patients were treated with FOLFIRINOX followed by SBRT [5]. The SBRT treatment was found to be safe, and the median overall survival (OS) in patients who received SBRT after FOLFIRINOX was 17 months (95% CI 14–21). As PDAC is considered an immunological cold tumor, the anti-tumor immune response in LAPC patients treated with SBRT monotherapy after systemic chemotherapy is probably not optimal. Adding an adjuvant to SBRT could improve the immunological conditions for an effective immune response. In this first-in-human trial, the addition of a vaccine containing a heat-killed mycobacterium obuense (IMM-101), to SBRT was investigated. IMM-101 has been demonstrated to induce the activation and maturation of dendritic cells in vitro [18]. Moreover, in a pancreatic cancer murine model, IMM-101 demonstrated to be able to produce protective CD8+ T cell responses [19]. A previous randomized controlled trial in patients with advanced pancreatic cancer investigated the value of adding IMM-101 to gemcitabine treatment [20]. The addition of IMM-101 to gemcitabine was associated with an improvement in OS from 4.4 to 7.0 months (95% CI 0.33–0.87, *p* = 0.01) in a pre-defined metastatic subgroup [20]. Next to this, an interesting case report presented a case of a patient with metastasized pancreatic cancer who underwent a synchronous resection of the primary tumor and liver metastases, after multimodality treatment with chemotherapy, IMM-101 and chemoradiation. This patient was free of disease four years after diagnosis [21]. Additionally, promising outcomes have been reported in melanoma patients treated with IMM-101 as well [22,23]. We hypothesize that IMM-101 vaccinations can enhance a host’s innate immune response, improving the immuno-modulatory effects and in situ vaccination efficacy of SBRT. 

In this study, we present the results of the immuno-monitoring of the peripheral blood in patients with locally advanced pancreatic cancer treated with SBRT and IMM-101, as well as their clinical outcome.

## 2. Treatment Scheme and Methods

### 2.1. Study Design and Participants

The LAPC-2 trial was a single-center, single-arm, non-randomized, open-label, phase I/II trial treating biopsy proven LAPC patients with SBRT and IMM-101, after prior treatment with at least 4 cycles of FOLFRINOX. LAPC was defined according to the guidelines of the Dutch Pancreatic Cancer Group as >90° contact with the superior mesenteric artery, the celiac axis and/or any hepatic artery and/or >270° contact with the superior mesenteric vein or the portal vein and/or occlusion of these veins [24]. Main inclusion criteria were (1) age > 18 years and < 75 years, (2) WHO performance status of 0 or 1, (3) normal renal and liver function, (4) largest tumor size <7 cm × 7 cm × 7 cm, and (5) no evidence of metastatic disease. Main exclusion criteria were (1) prior radiotherapy, chemotherapy other than FOLFIRINOX or pancreatic resection, (2) current or previous treatment with immunotherapeutic drugs, and (3) use of corticosteroids. The study was approved by the Central Committee on Research involving Human Subjects (NL68762.078.19) as defined by the Medical Research Involving Human Subjects Act. Procedures followed were in accordance with the ethical standards of these committees on human experimentation and with the Helsinki Declaration of 1975, as revised in 2008. The trial is registered with the Netherlands Trial Register, NL7578. Written informed consent was obtained from each subject. All detailed inclusion and exclusion criteria are listed in Appendix A.

### 2.2. SBRT and IMM-101 Vaccination

The tumors were irradiated with the Cyberknife (Accuray Incorporated, Sunnyvale, CA, USA). To accurately guide the radiation, the gastroenterologist placed three radio-opaque markers in or near the tumor (within 3cm of the tumor). Patients received a total of 40 Gray (Gy) of SBRT in five fractions on consecutive days. Radiation started at week 2, just after patients received the second vaccination of IMM-101. Immodulon Therapeutics Ltd. (Uxbridge, UK) produced and shipped pre-labeled IMM-101 vials to the pharmacy of the Erasmus MC University Medical Center. IMM-101 was injected intradermally over the deltoid muscle by the standard Mantoux intradermal injection technique. One mL was injected, which contained one milligram of IMM-101. IMM-101 was administered six times: i.e., on week 0, week 2, week 4, week 8, week 10 and week 12. Figure 1 illustrates the treatment schedule. At week 0, week 2, week 4, week 8 and week 14 blood draws were performed for immune-monitoring;, i.e., before planned study drug administration or SBRT treatment. One red 10 mL clot activator tube from BD Vacutainer^®^, one 3 mL Tempus^TM^ RNA stabilisator tube and two 10 mL EDTA tubes from BD Vacutainer^®^ were collected. The blood was processed within six hours after collection. Plasma, serum and peripheral blood mono-nuclear cells (PBMCs) were isolated and cryopreserved.

### 2.3. Follow up and Resectability Assessments

At week 14, resectability was assessed based on CT scans, biochemical response and the patients’ clinical situation. An explorative laparotomy was performed in fit patients with a possibly resectable tumor and a >50% decrease in CA 19.9. In case of local and distant tumor progression, the patient was referred to the medical oncologist. The decision for an explorative laparotomy was made by a multidisciplinary tumor board consisting of at least a radiologist specialized in abdominal radiology, an experienced pancreas surgeon and a medical oncologist. After completion of IMM-101 treatment, routine follow-up was started until the time of death or 5 years after completion of SBRT. Follow-up visits included regular CT scans and tumor-marker assessments.

### 2.4. Objectives and Endpoints

The primary objective of the phase I study was to determine the safety of adding IMM-101 to SBRT. The endpoint for this objective was an observed toxicity rate of grade 4 or higher related to the study treatment. Toxicities were scored according to CTCAE criteria version 5.0 [25]. The secondary objective was to investigate the immuno-modulatory effects of the combination treatment in the peripheral blood. Endpoints for this were the changes in the circulating immune cell compartment on RNA and protein level.

### 2.5. Targeted Gene-Expression Profiling

RNA was isolated from Tempus blood tubes using Tempus ^TM^ Spin RNA Isolation Reagent Kit (Thermo Fisher Scientific, Breda, The Netherlands). Isolated RNA was purified using RNeasy^®^ MinElute^®^ Cleanup Kit (Qiagen, Leiden, The Netherlands). The RNA quantity and quality were measured using the Agilent 2100 BioAnalyzer (Santa Clara, CA, USA). The RNA concentration was corrected to include the fragments ≥300 bp. For each sample, 200 ng of RNA was hybridized with probes of the PanCancer Immune profiling panel (730 innate and adaptive immune related genes and 40 housekeeping genes) for 17 h at 65 °C, following the manufacturing procedure (NanoString Technologies Inc., Seattle, WA, USA). The nCounter^®^ FLEX platform was used to wash the extra probes, and genes were counted by scanning 490 Fields-of-view (FOV). The raw data of gene counts were uploaded to the nSolver™ Data Analysis software (version 4.0, NanoString, Seattle, WA, USA). The gene counts were normalized using the Advanced Analysis module (version 2.0) of nSolver™.

### 2.6. Flow Cytometry Immuno-Monitoring

For the enumeration of immune subsets, whole blood was freshly stained for flow cytometry. In addition, longitudinal immuno-monitoring was performed on liquid nitrogen stored PBMCs. Cell surface staining was carried out after blocking Fc receptors by incubating cells with fluorescently conjugated mAbs directed against CD4 (SK3), CD11b (ICRF44), CD14 (M5E2), CD19 (HIB19), CD20 (2H7) CD56 (NCAM16.2), CD86 (FUN-1), HLA-DR (G46-6), ICOS (DX29) and ICOS-L (2D3/B7-H2) (all BD Biosciences, Erebodegem, België); CD8 (SK1), CD11c (BV605), CD15 (HI98), CCR7 (G043H7), LAG-3 (11C3C65), PD-1 (EH12.2H7), TIM-3 (F38-2E2) (all BioLegend, Amsterdam, The Netherlands); and CD3 (UCHT1), CD33 (WM-53), CD45RA (MEM-56), CTLA-4 (14D3), FOXP3 (236A/E7), Ki-67 (20Raj1) (all Thermo Fisher Scientific). Intracellular transcription factor staining was performed using the FoxP3 Staining Buffer Set (Thermo Fisher Scientific). Cells were in addition stained for viability using fixable LIVE/DEAD aqua cell stain (Thermo Fisher Scientific). Data were acquired using the Symphony flow cytometer (BD Biosciences) and analyzed with FlowJo v10.7. Cell subsets are gated as previously described [26,27].

### 2.7. Statistical Analysis—Sample Size Calculation

The primary objective of the phase I trial was to determine the safety of adding IMM-101 to SBRT. In our previous LAPC-1 trial, the grade 4 toxicity rate of SBRT was 10% [5]. With a sample size of 20 for the phase I trial, we were able to estimate a toxicity rate of 10% within a 95% confidence interval of [1.2–31.7%] using the binomial exact method. This means that a maximum of 6/20 (30%) patients were allowed to have grade 4 toxicity or higher for the treatment to be regarded as safe and before proceeding to the phase II trial.

### 2.8. Statistical Analysis—Data Analysis and Visualisation

Baseline patient characteristics are summarized using the median and interquartile range for continuous variables and using counts and percentages for categorical variables. PFS and OS were calculated from start date of FOLFIRINOX chemotherapy to the first documented event. Survival estimates were calculated using Kaplan–Meier method. Flow cytometry data were normalized for baseline. Paired Wilcoxon signed-ranks tests were used to test for significance between baseline measurements and other timepoints. Figures were made using GraphPad Prism software v8.0. Gene-expression data were corrected for multiple testing using the Benjamini–Hochberg procedure. In all cases, a *p*-value of 0.05 and below was considered significant (*), *p* < 0.01(**) and *p* < 0.001 (***) as highly significant. The heat map was generated using the average log2 normalized gene expression of the significant differentially expressed genes per week. The heat map was visualized using the web-based tool Morpheus [28]. The Spearman correlations were calculated using the PFS or OS and the absolute difference between baseline and week 4 (after IMM101/SBRT) of activated cell frequencies. The volcano plots and correlations were visualized in R (version 4.1.1).

## 3. Results

### 3.1. Patient and Treatment Characteristics

A total of 21 patients were included in the phase I, LAPC-2 trial, between October 2019 and June 2020. The first included patient (IMM001) had a liver metastasis, which was found during endoscopic ultrasound that was performed to place the radio-opaque markers for the SBRT. This patient was, therefore, excluded. Eventually, 20 patients received study treatment. Patients were treated with a median of 8 (8–9) cycles of FOLFIRINOX before inclusion in the trial. The median time between FOLFIRINOX and the first IMM-101 vaccination was 6.4 (5.2–7.8) weeks. The median age was 63 (60–68) years and 11 (55%) were male. Their median body mass index was 24 (21–28) kg/m^2^. All patients received the total dose of 40 Gy of SBRT. Nineteen patients received the six planned vaccinations with IMM-101 and one patient received only three vaccinations due to disease progression. Immune analyses of the PBMCs were performed in 19/20 patients due to the absence of sufficient PBMCs in patient IMM016. Gene expression analyses were performed in 19/20 patients because we were not able to isolate RNA from IMM017. Detailed patient and treatment characteristics are shown in Table 1.

### 3.2. Safety and Clinical Outcome

In 6/20 patients, we observed eleven grade 3 adverse events, of which three were considered to be possibly related to SBRT. None were related to IMM-101. Toxicity of grade 4 or higher was not observed. All patients experienced mild injection-site reactions, ranging from erythema to skin abscesses, with none resulting in systemic symptoms. Table 2 shows all grade 3 or higher toxicities. At present, (i.e., May 2022), 18/20 (90%) patients have experienced progression (local or distant) of disease and 17/20 (85%) patients have died. In all patients, the median PFS was 11.7 months (95% CI: 10.2–13.3) and the median OS was 17.8 months (95% CI: 11.3–24.4). The median PFS and median OS of the unresected patients (n = 16) was 11.2 (95% CI: 8.0–14.4) and 17.8 (95% CI: 12.0–23.6) months, respectively. Four (20%) patients underwent a resection of the tumor. In one patient, a small, solitary liver metastasis was found during explorative laparotomy and the primary tumor and the metastasis were both resected. This patient was free of disease 15 months after the operation. Another patient experienced local recurrence of disease four months after the resection. This was treated with systemic chemotherapy. In the absence of disease progression, a re-resection was performed 12 months after the initial resections. This patient was free of disease 8 months after the re-resection. Two patients died from complications from the operation.

### 3.3. Downregulation of Genes Related to Lymphocyte Subsets and Immune inhibition after IMM-101/SBRT

Targeted gene expression profiling was performed to investigate the effect of IMM-101 and SBRT on the immune cells. Apart from increased expression of three genes (i.e., LTF, CAMP and LCN2) at baseline, no significant differences were observed between baseline (week 0) and after one vaccination IMM-101 (week 2) (Appendix A). However, in week 4, after SBRT combined with IMM-101, profound changes were observed in immune-related gene expression (Figure 2A,B). Various genes related to lymphocyte subsets were downregulated (i.e., CD8a, MS4A1, CD22, CD79A, KLR family genes). Furthermore, genes related to lymphocyte inhibition/exhaustion (i.e., BTLA, TBX21, KLRC1) were also downregulated after IMM101/SBRT treatment. These results indicate changes in the circulating lymphoid compartment of LAPC patients specifically after combined IMM-101/SBRT treatment.

### 3.4. Reduced Peripheral Lymphocyte Numbers following IMM-101/SBRT

We additionally assessed various immune subsets in the peripheral blood using flow cytometry. No significant changes in immune subsets were observed two weeks after the first vaccination with IMM101. The addition of SBRT transiently reduced CD4+ and CD8+ T cells, CD19+ B-lymphocytes and CD56+ NK cells (Figure 3). SBRT did not curtail the myeloid compartment (i.e., CD15+CD16− eosinophils, CD15+CD16+ neutrophils, CD14+CD16− monocytes, CD14−CD16−CD11c+ dendritic cells). Additionally, the number of CD14+CD16−CD11b+HLA-DR^low^ MDSCs increased after combining SBRT and IMM-101 (Appendix A). Lymphocyte cell numbers recovered at week 8, within 6 weeks after SBRT.

### 3.5. IMM-101/SBRT Increased Proportions of Activated Lymphocytes

In-depth longitudinal immune monitoring was performed to further describe the phenotypic characteristics of immune cells following study therapy. We did not find changes in activation or inhibitory marker expression on CD4+ regulatory T cells or CD4+ T helper cells or cytotoxic CD8+ T cells after one vaccination with IMM-101 in week 2. In contrast, the addition of SBRT significantly increased the frequencies of activated CD4+ and CD8+ T cells and CD56+ NK cells in week 4 as indicated by the markers ICOS, HLA-DR as well as the combined increase in Ki67 and PD-1 levels. Notably, this increase was not observed for the inhibitory markers PD-1, TIM-3 and LAG-3, although we did observe significantly upregulated CTLA-4 levels on the CD4+ Non-Tregs after combination therapy. Furthermore, the increase in activated CD4+ and CD8+ T cell frequencies was mainly driven by the memory compartment (i.e., CCR7+CD45RA− central memory and CCR7−CD45RA− effector memory). One vaccination of IMM-101 did significantly increase activated CD86+CD19+ B cell frequencies in week 2. The addition of SBRT further activated these CD19+ B cells demonstrated by increased Ki67+PD-1+ and CD86+ frequencies. Lastly, IMM-101/SBRT transiently induced higher frequencies of CD11c+ dendritic cells, HLA-DR+CD14+ macrophages and HLA-DR−CD14−CD15− DN-MDSCs. Data are shown in detail in Figure 4 and Appendix A.

### 3.6. Treatment-Induced Increase in Activated Lymphocytes Is Correlated with Survival

To explore if treatment-induced effects could be translated to clinical outcome, we analyzed if absolute differences in immune cell status between treatment-naïve (week 0) and study treatment samples (week 4) were correlated with survival. Patients who underwent a resection (n = 4) were excluded from this analysis, since a resection possibly influences PFS and OS outcomes. Another patient (IMM016) was excluded from the analysis due to an absence of sufficient PBMCs. Therefore, eventually 15 patients were included in the analysis. We found that increased levels of CD28+ effector memory (CCR7−CD45RA+) cytotoxic T cells correlated with improved PFS and OS (Figure 5).

## 4. Discussion

In this first-in-human trial, we firstly assessed the safety of IMM101/SBRT treatment, in patients with LAPC after prior treatment with FOLFIRINOX chemotherapy.

All patients experienced injection site reactions, which were uncomfortable for some patients. Eleven grade 3 toxicities were observed, of which three were possibly related to SBRT treatment. No grade 4 or higher toxicities were reported and none of the observed toxicities were considered to be related to IMM-101. This treatment approach demonstrated to be safe, and the trial proceeded to the phase II trial.

Secondly, we investigated the immune-modulatory effects of IMM-101/SBRT treatment in the peripheral blood. Two weeks after the first vaccination with IMM-101, no explicit changes on gene expression and protein level in the immune system of LAPC patients could be demonstrated. After treatment with IMM-101 with SBRT, we observed a downregulation of genes related to lymphocyte subsets, and this lymphodepletion was confirmed by flow cytometry. Interestingly, IMM-101/SBRT treatment did induce a rise in the number of MDSCs. Radiotherapy-induced MDSC expansion in patients with PDAC has previously been described [29]. It is also likely that SBRT and not IMM-101 induced the lymphodepletion, seeing that, in a previous study, external beam radiotherapy caused systemic immune-cell depletion [30]. Except MDSCs, cell numbers of other cell subsets within the myeloid compartment did not significantly increase. The latter may be explained by the fact that the radioresistance of suppressive myeloid cells is stronger than that of lymphocytes [31].

Our combined gene expression and flow cytometry analyses demonstrated therapy-induced activation of T cell and NK cell subsets, with no increase in most inhibitory markers (i.e., PD-1, TIM-3 and LAG-3). Interestingly, therapy-induced activation of T cells occurred mainly in the memory compartment, which may be beneficial for seeding the tumor with antigen-specific T cells to mount successful anti-tumor responses. In agreement with this notion, improved PFS and OS were correlated with increased levels of activated effector memory cytotoxic T cells. In pre-clinical models, ablative doses of radiotherapy have been associated with improved intratumoral CD8+ T cell infiltration due to increased antigenicity of malignant cells, or by promoting immuno-stimulatory signals to recruit and activate antigen-presenting cells [32,33].

We found limited significant changes 2 weeks after the first vaccination with IMM-101. Still, the CD86+ expression on B cells increased. Adding SBRT further augmented the B cell activation, as demonstrated by the increase in Ki67+PD-1+ and CD86+ frequencies. A higher B cell activation may be beneficial, as B cell activation has been associated with positive responses to cancer-immunotherapy [34,35].

SBRT may hypothetically improve the anti-tumor efficacy of IMM-101 through antigen release upon tumor destruction, inducing in situ vaccination. IMM-101 could concurrently provide enhanced innate immunity to engage robust T cell responses. Unfortunately, the current study design did not allow for us to investigate this mechanism. Next to this, the common limitations of phase I/II trials, such as a small sample size and the lack of a control group, also applied to this study. However, the sample size was adequate to prove the safety of the combination treatment. Moreover, despite the low number of patients, a clear trend in immunological changes could be observed in most patients, which strengthens the hypothesis that treatment-induced immune modulation existed. Due to the lack of a control group, the observed changes could theoretically be better explained by time than by a cause–effect phenomenon caused by the treatment. However, certain factors argue against this. Firstly, between week 0 and 2, no significant changes occurred. In contrast, between week 2 and week 4, drastic changes were observed in the peripheral immunity. This occurred after the second vaccination and the SBRT treatment. The lack of changes in the first two weeks, compared to the extensive changes that occurred between week two and four, combined with the timing of treatment, argue against the hypothesis that the immunological changes were mostly impacted by time. Secondly, the observed immunological changes after SBRT/IMM-101 treatment tended to restore mostly to baseline after time progressed. If time and, thus, disease progression was the main factor explaining the changes in the immune system, one would expect these changes to persist as time progressed. Another limitation of this study is that our analysis was only focused on peripheral immunity. A local assessment of the immune composition would have improved understanding of the study–treatment effect, as SBRT acts directly on the tumor. Nonetheless, the upregulation of immune checkpoints on circulating T cells, including CTLA-4, endorse the addition of immune-checkpoint blocking antibodies in future studies. Moreover, combining checkpoint-blocking antibodies with radiotherapy alone, or possibly with IMM-101, has shown promising results in pre-clinical models [36,37]. In addition to combination with immune-checkpoint-blocking antibodies, intratumoral administration of IMM-101 could improve its clinical efficacy. The most-used mycobacterium vaccine is the live-attentuated *Mycobacterium Bovis* Bacillus Calmette-Guérin (BCG) vaccine [38]. This tuberculosis vaccine was demonstrated to be able to induce potent anti-tumor immunity and adjuvant intravesical BCG instillations after a transurethral resection of bladder cancer, and was proved to be effective in preventing bladder cancer recurrence [39,40,41,42,43]. The administration of the vaccine at the disease site might be important to its efficacy.

## 5. Conclusions

In this open-label, single-center, phase I study, the safety and immunomodulatory effects of intradermal IMM-101 with SBRT were investigated in patients with LAPC. We observed transient lymphodepletion and enhanced T cell activation in the peripheral blood. Increased levels of activated T cells after treatment correlated with improved PFS and OS. Future studies are needed to provide mechanistic insights into how these observations are linked to clinical efficacy. The intratumoral administration of IMM-101 and combinations with other immunotherapeutic agents focusing on adaptive responses (e.g., immune checkpoint blockade, adoptive cell transfer therapy) may lead to improved efficacy for this group of patients with limited treatment options.

## Figures and Tables

**Figure 1 cancers-14-05299-f001:**
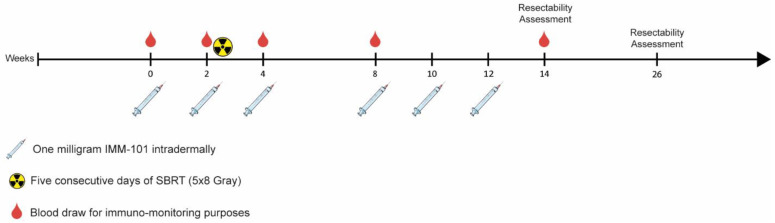
Schematic treatment schedule. After discontinuation of FOLFIRINOX treatment, patients were included in the trial. Patients received three bi-weekly intradermal vaccinations of IMM-101 at weeks 0, 2 and 4. At week 2, after the second vaccination, stereotactic body radiotherapy (SBRT) treatment started. Patients received 5 × 8 Gy of SBRT. They received three more vaccinations at weeks 8, 10 and 12. At week 14, the first resectability assessments was performed. Some patients were offered an explorative laparotomy with possible resection.

**Figure 2 cancers-14-05299-f002:**
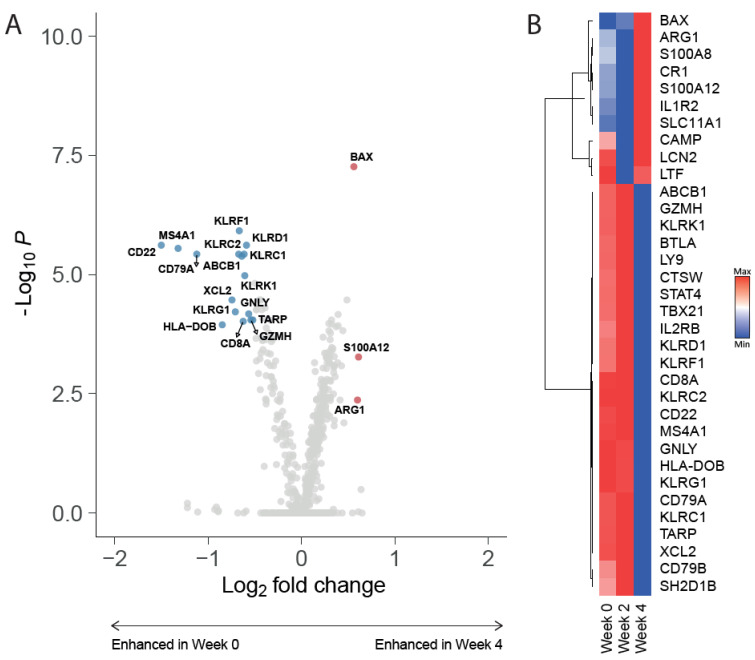
SBRT/IMM-101 induced gene expression. (**A**) Volcano plot demonstrating genes upregulated at baseline versus week 4. Highlighted genes underwent a log_2_fold change < −0.5 or >0.5 and *p*-value < 0.05. (**B**) Heat map of significantly differentially expressed genes between week 0, week 2 and week 4.

**Figure 3 cancers-14-05299-f003:**
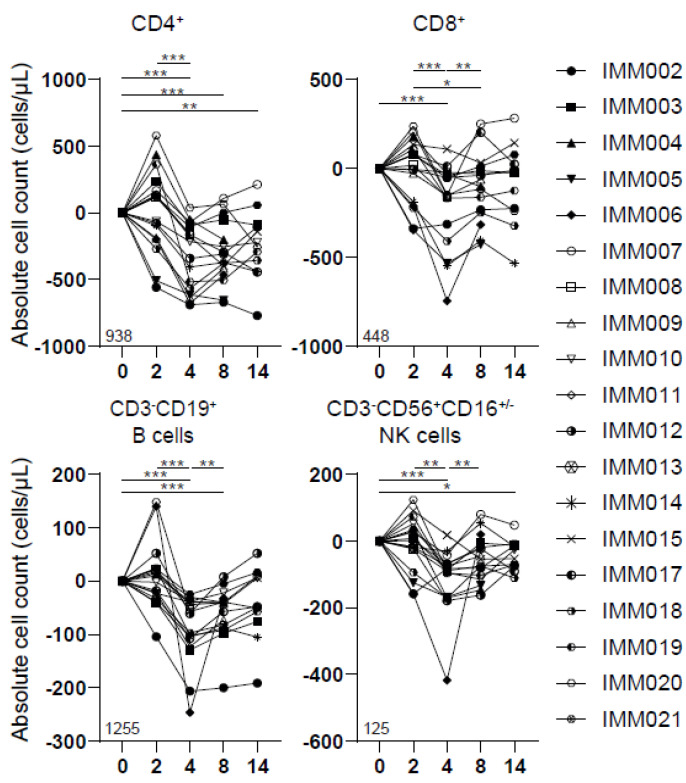
SBRT/IMM-101 induced transient lymphodepletion. Number of CD4+, CD8+, CD3−CD19+ and CD3−CD56+CD16+/−peripheral blood lymphocytes per µL blood. N = 19. Data were normalized for baseline (week 0) and paired per patient. Percentage in the bottom left corner is the average frequency at baseline. Significance was determined using the paired Wilcoxon signed-rank test. * *p* < 0.05, ** *p* < 0.01, *** *p* < 0.0001.

**Figure 4 cancers-14-05299-f004:**
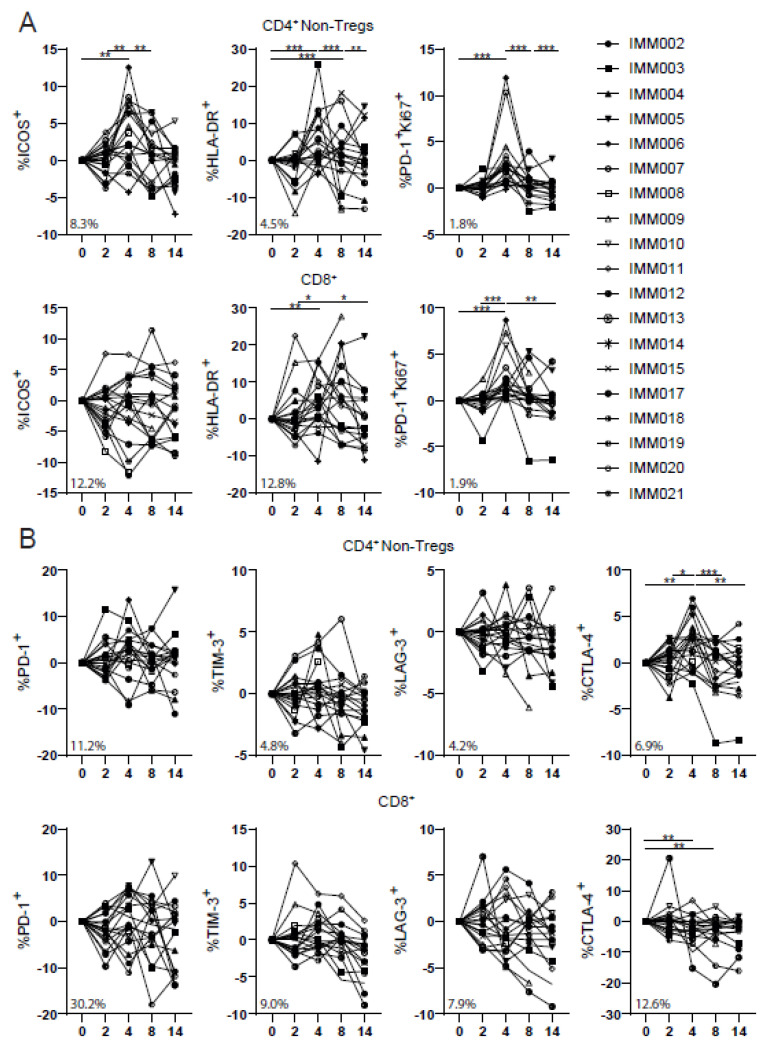
SBRT/IMM-101 induced T-cell activation. (**A**) Percentage of ICOS+, HLA-DR+, PD-1+/Ki67+ subsets of CD4+ Non-Tregs and CD8+ cells. (**B**) Percentage of PD-1+, TIM-3+, LAG-3, CTLA-4+ subsets of CD4+ Non-Tregs and CD8+ cells. N = 19. Data were normalized for baseline (week 0) and paired per patient. Percentage in the bottom left corner is the average frequency at baseline. Significance was determined using the paired Wilcoxon signed-rank test. * *p* < 0.05, ** *p* < 0.01, *** *p* < 0.0001.

**Figure 5 cancers-14-05299-f005:**
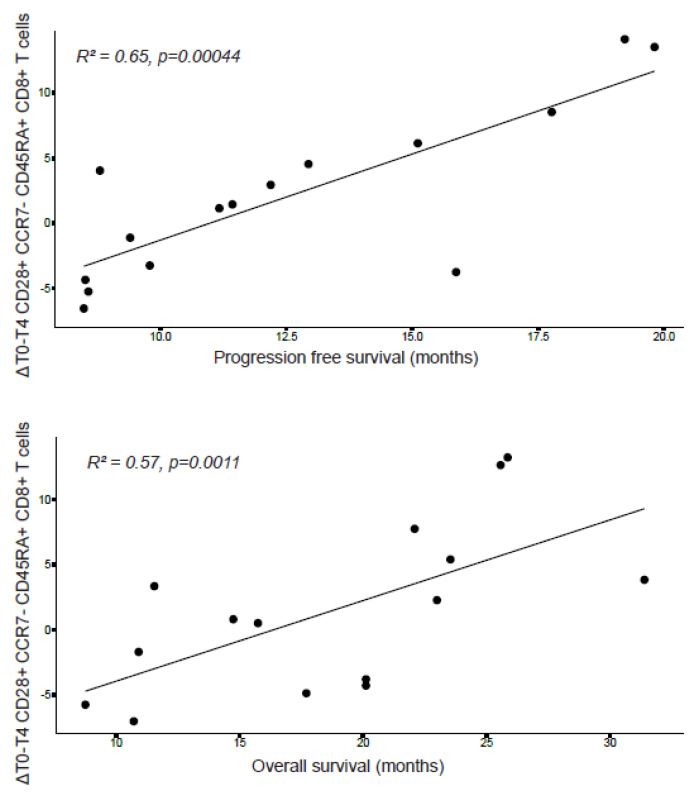
Treatment-induced T-cell activation correlated with improved progression-free survival. Spearman correlation plots demonstrating a positive correlation between IMM101/SBRT-induced absolute difference of CD28+ CCR7− CD54RA+ cytotoxic T cells and progression-free survival and overall survival. N = 15.

**Table 1 cancers-14-05299-t001:** Patient and treatment characteristics.

Patient Characteristics	N = 20 (IQR) or [%]
Age, years	63 (60–68)
Male sex	11 [55]
BMI, kg/m²	24 (21–28)
ECOG performance status *	
0	4 [20]
1	16 [80]
CA 19.9 at inclusion, kU/L	101 (43–137)
CEA at inclusion, µg/L	4.4 (3.5–5.8)
Leukocyte count at inclusion, ×10^9^/L	6.7 (4.7–9.9)
Platelet count at inclusion, ×10^9^/L	195 (133–232)
Neutrophil count at inclusion, ×10^9^/L	3.6 (2.7–7.2)
Lymphocyte count at inclusion, ×10^9^/L	1.4 (1.2–1.8)
SII, (N x P) / L	624 (311–889)
NLR	3.1 (2.3–5.0)
PLR	147 (87–171)
**Treatment characteristics**	
Biliary stenting at diagnosis	9 [45]
Diagnostic laparoscopy at diagnosis	6 [30]
FOLFIRINOX treatment	20 [100]
FOLFIRINOX, cycles	8 (8–9)
Interval stop FOLFIRINOX and start IMM-101, weeks	6.4 (5.2–7.8)
40 Gray of SBRT	20 [100]
IMM-101	20 [100]
Six vaccinations	19 [95]
Three vaccinations	1 [5]
Resection	4 [20]

Statistics: Continuous variables are shown as medians with interquartile range and categorical variables are shown as counts with percentages. Abbreviations: BMI = body mass index, ECOG = Eastern Cooperative Oncology Group, CA 19.9 = carbohydrate antigen 19.9, CEA = carcinoembryonic antigen, SII = Systemic-Immune-Inflammation index, NLR = neutrophil to lymphocyte ratio, PLR = platelet to lymphocyte ratio, SBRT = stereotactic body radiotherapy, N = neutrophils, P = platelets, L = lymphocytes. * ECOG performance status 0 = Fully active, able to carry on all pre-disease performance without restriction. ECOG performance status 1 = Restricted in physically strenuous activity but ambulatory and able to carry out work of a light or sedentary nature, e.g., light housework, office work.

**Table 2 cancers-14-05299-t002:** Grade 3 or higher adverse events.

Subject	Adverse Event Term	Grade	Relation to SBRT	Relation to IMM-101
IMM003	Gastro-intestinal haemorrhage	3	Possibly	Unrelated
IMM006	Gastro-intestinal haemorrhage	3	Possibly	Unrelated
IMM007	Gastro-intestinal haemorrhage	3	Unrelated	Unrelated
IMM007	Gastro-intestinal haemorrhage	3	Possibly	Unrelated
IMM007	Stent disfunction	3	Unrelated	Unrelated
IMM007	Cholangitis	3	Unrelated	Unrelated
IMM007	Stent disfunction	3	Unrelated	Unrelated
IMM008	Cholestatis	3	Unrelated	Unrelated
IMM008	Cholangiosepsis	3	Unrelated	Unrelated
IMM009	Vertigo	3	Unrelated	Unrelated
IMM014	Duodenal obstruction	3	Unrelated	Unrelated

Toxicities were scored according to Common Terminology Criteria for Adverse Events (CTCAE) version 5.0 [25]. The treating physicians judged the possibility of a relation to the study treatment. Adverse events not related to SBRT or IMM-101 were considered to be related to pancreatic ductal adenocarcinoma. Abbreviations: SBRT = stereotactic body radiotherapy.

## Data Availability

All data, analytic methods, and study materials not presented in this manuscript or data repository, will be made available to other researchers upon reasonable request.

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
