# Peer review of "Immunomodulatory Effects of Stereotactic Body Radiotherapy and Vaccination with Heat-Killed Mycobacterium Obuense (IMM-101) in Patients with Locally Advanced Pancreatic Cancer"

_cancers, 2022, doi:10.3390/cancers14215299_

Round 1

Reviewer 1 Report

F.R. van t Land et al presented a first-in-human trial assessing the safety of IMM101/SBRT 367 treatment, in patients with LAPC after prior treatment with FOLFIRINOX. The study is focusing on a very interesting topic. Such an issue certainly deserves further evidence and confirmation in the next future. The study is well-designed, and the results are clear and well-explained. I have just two minor comments:

1) Among the limitations of the study it should be acknowledged the low number of patients, which impacts the drawing of any definitive conclusion regarding both the primary and the secondary endpoints of the study.

2) Authors should further clarify the indications for surgery after neoadjuvant FOLFIRINOX. In other words, how were patients selected for surgical exploration? Were all patients without local or distant progression explored for surgical resection?

3) The current work is suffering from the lack of a real control group. The association between SBRT/IMM-101 and immunological changes might have been impacted by the time more than by a cause-effect phenomenon. How authors are going to deal with that? The authors should comment on that. 

Author Response

Point-by-point reply

Reviewer 1

F.R. van t Land et al presented a first-in-human trial assessing the safety of IMM101/SBRT 367 treatment, in patients with LAPC after prior treatment with FOLFIRINOX. The study is focusing on a very interesting topic. Such an issue certainly deserves further evidence and confirmation in the next future. The study is well-designed, and the results are clear and well-explained. I have just two minor comments:

Comment 1: Among the limitations of the study it should be acknowledged the low number of patients, which impacts the drawing of any definitive conclusion regarding both the primary and the secondary endpoints of the study.

Reply to comment 1 of first reviewer: Dear Mr/Mrs, thank you very much for this comment. Indeed, we agree that the low sample size, a common limitation of phase I/II trials, also applies to the current study. An increase in the sample size would possibly strengthen the conclusions to the primary and secondary endpoints. This was added to the discussion of the manuscript. Despite the fact that this was a limitation, we do think the sample size was adequate to prove safety of the combination treatment. Moreover, as we observed a clear trend in immunological changes in most patients, we expect that the concept of treatment induced immune modulation existed.

Comment 2: Authors should further clarify the indications for surgery after neoadjuvant FOLFIRINOX. In other words, how were patients selected for surgical exploration? Were all patients without local or distant progression explored for surgical resection?

Reply to comment 2 of first reviewer: Dear Mr/Mrs, thank you for this question. In section 2.3. Follow up and resectability assessments, this was already explained briefly: “An explorative laparotomy was performed in fit patients with a possibly resectable tumor and a >50% decrease of CA 19.9. In case of local and distant tumor progression, the patient was referred to the medical oncologist.” The following was added to this section of the manuscript:” The decision for an explorative laparotomy was made by a multidisciplinary tumor board consisting of at least a radiologist specialized in abdominal radiology, an experienced pancreas surgeon and a medical oncologist”.

Comment 3: The current work is suffering from the lack of a real control group. The association between SBRT/IMM-101 and immunological changes might have been impacted by the time more than by a cause-effect phenomenon. How authors are going to deal with that? The authors should comment on that.

Reply to comment 3 of the first reviewer: Dear Mr/Mrs, thank you for the valid comment. Indeed, the lack of a control group is a limitation. This limitation is common to phase I/II trials, and implies that the association between the treatment and immunological changes could have been impacted significantly by time, more than by a cause-effect phenomenon of the treatment. However, certain factors argue against this. Firstly, between week 0 and 2, no significant changes occur. In contrast, between week 2 and week 4, drastic changes in the peripheral immunity are observed. This is after the second vaccination and the SBRT treatment. The lack of changes in the first two weeks, compared to the extensive changes between week two and four, combined with the timing of treatment, argue against the hypothesis that the immunological changes were impacted mostly by time.  Secondly, the observed immunological changes after SBRT/IMM-101 treatment, tend to restore mostly to baseline after time progresses. If time, and thus disease progression, was the factor explaining the changes in the immune system, one would expect that these changes would persist as time progresses.

Reviewer 2 Report

PDAC has such a dismal outcome that any trial addressed to improve treatment results is welcome.

More than 3300 clinical trials in the last 15 years registered in clinicaltrials.gov are a strong witness to the unsolved problem.

The authors propose an immunological treatment associated to radiotherapy and chemotherapy.

The idea is not new but it has not been thoroughly investigated in PDAC. This approach is a valid and needed research at this stage of failed treatments.

The authors performed a limited scope research on the issue and arrived to promising results that will require further research. It is a first step in the right direction.

However, there are some observations on the paper.

Lines 64/65 PDAC treatment preference is not limited to FOLFIRINOX. The reference they give, the NCCN guidelines clearly includes the gemcitabine plus nab-paclitaxel as the first line of treatment. Actually many treatment centers use the gemcitabine-nab paclitaxel association as the gold standard because it is much better tolerated by patients.

Please add in the paper the gemcitabine-nab paclitaxel.

The statistical  analysis presented in the paper is biased. It includes in the same group those cases in which the treatment was neoadjuvant, 4 cases, and those in which it was adjuvant. These two groups usually have different PFS and OS.  I believe that these four cases should be excluded in the PFS and OS consideration.

However, the paper is not addressed to show results because this is a phase I study and therefore the bias mentioned above does not change the toxicity exploration.

Figures 3 and 4 are impossible to be analyzed in the way they are presented. The best thing the authors can do is to eliminate them.

In Figure 5 the overall survival and PFS calculations are based on 14 patients. That means that six patients have been excluded. The authors should explain who and why was excluded.

There are important publications that were omitted in the reference and the discussion, Including some of them would enrich the discussion. Examples of references not mentioned:

Bilyard, H., Mines, C., Brunet, L.R. et al. IMM-101, an immunotherapeutic agent in clinical development as an adjunctive treatment for pancreatic cancer. j. immunotherapy cancer 2 (Suppl 3), P83 (2014). https://doi.org/10.1186/2051-1426-2-S3-P83

Elia, A., Lincoln, L., Brunet, L. R., & Hagemann, T. (2013). Treatment with IMM-101 induces protective CD8+ T cell responses in clinically relevant models of pancreatic cancer. Journal for ImmunoTherapy of Cancer1(1), 1-1. https://doi.org/10.1186/2051-1426-1-S1-P215

Neves, M. C., Giakoustidis, A., Stamp, G., Gaya, A., & Mudan, S. (2015). Extended survival after complete pathological response in metastatic pancreatic ductal adenocarcinoma following induction chemotherapy, chemoradiotherapy, and a novel immunotherapy agent, IMM-101. Cureus7(12). doi:10.7759/cureus.435

Crooks, J., Brown, S., Gauthier, A., de Boisferon, M. H., MacDonald, A., & Brunet, L. R. (2016). The effects of combination treatment of IMM-101, a heat-killed whole cell preparation of Mycobacterium obuense (NCTC 13365) with checkpoint inhibitors in pre-clinical models. Cell10, 20. https://www.immodulon.com/wp-content/uploads/2018/09/SITC-Immodulon-2016-POSTER-FINAL-002.pdf

Fowler, D., Dalgleish, A., & Liu, W. (2014, November). A heat-killed preparation of Mycobacterium obuense can reduce metastatic burden in vivo. In Journal for ImmunoTherapy of Cancer (Vol. 2, No. Suppl 3). BMJ Publishing Group LTD. DOI:10.1186/2051-1426-2-S3-P54

Brunet, L. R., Carroll, K., Summerton, L., Wagle, S., & Ducreux, M. (2017). Baseline carcinoembryonic antigen predicts response to IMM-101 in advanced pancreatic cancer: Data analysis from a randomised, open label, phase II study. Annals of Oncology28, iii70-iii71.

Dalgleish, A.G., & Dalgleish, A.G. (2022). The role of immune modulation and anti‑inflammatory agents in the management of prostate cancer: A case report of six patients. Oncology Letters, 24, 247. https://doi.org/10.3892/ol.2022.13367

Author Response

Point-by-point reply

Reviewer 2

PDAC has such a dismal outcome that any trial addressed to improve treatment results is welcome. More than 3300 clinical trials in the last 15 years registered in clinicaltrials.gov are a strong witness to the unsolved problem. The authors propose an immunological treatment associated to radiotherapy and chemotherapy. The idea is not new but it has not been thoroughly investigated in PDAC. This approach is a valid and needed research at this stage of failed treatments. The authors performed a limited scope research on the issue and arrived to promising results that will require further research. It is a first step in the right direction. However, there are some observations on the paper.

Comment 1: Lines 64/65 PDAC treatment preference is not limited to FOLFIRINOX. The reference they give, the NCCN guidelines clearly includes the gemcitabine plus nab-paclitaxel as the first line of treatment. Actually many treatment centers use the gemcitabine-nab paclitaxel association as the gold standard because it is much better tolerated by patients. Please add in the paper the gemcitabine-nab paclitaxel.

Reply to comment 1 of the second reviewer: Dear Mr/Mrs, firstly we would like to thank the reviewer for reviewing and commenting on the manuscript. This is highly appreciated. Thank you for this valid comment. Indeed, gemcitabine with nab-paclitaxel is a good option as first line treatment, with indeed is usually (not always), better tolerated than FOLFIRINOX. We added gemcitabine with nab-paclitaxel as a first line treatment option for locally advanced pancreatic cancer patients to the introduction section of the manuscript.

Comment 2: The statistical analysis presented in the paper is biased. It includes in the same group those cases in which the treatment was neoadjuvant, 4 cases, and those in which it was adjuvant. These two groups usually have different PFS and OS.  I believe that these four cases should be excluded in the PFS and OS consideration. However, the paper is not addressed to show results because this is a phase I study and therefore the bias mentioned above does not change the toxicity exploration.

Reply to comment 2 of the second reviewer: Dear Mr/Mrs, thank you for your comments. We assume that this comment applies to figure 5, in which we correlate the progression free survival with the immune activation upon treatment. Indeed, we agree that a resection of the tumor, can possibly influence the progression free survival. Therefore, the statistical correlation between immune-activation and PFS could be influenced by the four cases who underwent a resection of the tumor. The correlation plots were made excluding the four patients who underwent a resection. Moreover, we have calculated the median RFS of the unresected subgroup of patients and added this to section 3.2 Safety and clinical outcome. As mentioned by the reviewer, indeed the resections did not influence the conclusions made regarding the primary outcome (i.e., the safety exploration).  

Comment 3: Figures 3 and 4 are impossible to be analyzed in the way they are presented. The best thing the authors can do is to eliminate them.

Reply to comment 3 of the second reviewer: Dear Mr/Mrs, thank you for this question. This is a valid point. Because we attempted to show most of the data in these figures, they become hard to read. Therefore, we decided to change these figures, and included the most important results, which were also highlighted in the results section of the manuscript. The rest of the data was added as a supplementary figure. In this way, readers can review the rest of the data if they want to. We do think it is important to visualize the longitudinal immunological changes. These figures are not created, for the reader to analyze the longitudinal trajectory of an individual patient. However, the figures do clearly show the trend in immunological changes for the whole group.

Comment 4: In Figure 5 the overall survival and PFS calculations are based on 14 patients. That means that six patients have been excluded. The authors should explain who and why was excluded.

Reply to comment 4 of the second reviewer: Dear Mr/Mrs, thank you very much for this question. Because of the comment we have found an error in the data computing the figures. On accident, three cases were excluded that should not have been excluded. For this analysis, two of the operated patients were excluded because they died from complications of the operation, and one patient was excluded because there were no sufficient PBMCs available for this patient. This resulted in a figure with n=14 patients. We did the analysis again, with the correct patients. We excluded 4 patients who underwent a resection, as this possibly influenced the PFS outcome, as advised by the reviewer. Moreover, one patient was excluded from the analysis because there were no sufficient PBMCs available (subject IMM016). This resulted in an analysis with n=15 patients. The figure now shows a positive correlation between an absolute increase in CD28+ CCR7- CD54RA+ cytotoxic T cells and progression free-survival and overall survival. The new analysis changed the results for supplementary figure 3. Because of this supplementary figure 3 was removed.

Comment 5

There are important publications that were omitted in the reference and the discussion, Including some of them would enrich the discussion. Examples of references not mentioned:

Bilyard, H., Mines, C., Brunet, L.R. et al. IMM-101, an immunotherapeutic agent in clinical development as an adjunctive treatment for pancreatic cancer. j. immunotherapy cancer 2 (Suppl 3), P83 (2014). https://doi.org/10.1186/2051-1426-2-S3-P83

Elia, A., Lincoln, L., Brunet, L. R., & Hagemann, T. (2013). Treatment with IMM-101 induces protective CD8+ T cell responses in clinically relevant models of pancreatic cancer. Journal for ImmunoTherapy of Cancer1(1), 1-1. https://doi.org/10.1186/2051-1426-1-S1-P215

Neves, M. C., Giakoustidis, A., Stamp, G., Gaya, A., & Mudan, S. (2015). Extended survival after complete pathological response in metastatic pancreatic ductal adenocarcinoma following induction chemotherapy, chemoradiotherapy, and a novel immunotherapy agent, IMM-101. Cureus7(12). doi:10.7759/cureus.435

Crooks, J., Brown, S., Gauthier, A., de Boisferon, M. H., MacDonald, A., & Brunet, L. R. (2016). The effects of combination treatment of IMM-101, a heat-killed whole cell preparation of Mycobacterium obuense (NCTC 13365) with checkpoint inhibitors in pre-clinical models. Cell10, 20. https://www.immodulon.com/wp-content/uploads/2018/09/SITC-Immodulon-2016-POSTER-FINAL-002.pdf

Fowler, D., Dalgleish, A., & Liu, W. (2014, November). A heat-killed preparation of Mycobacterium obuense can reduce metastatic burden in vivo. In Journal for ImmunoTherapy of Cancer (Vol. 2, No. Suppl 3). BMJ Publishing Group LTD. DOI:10.1186/2051-1426-2-S3-P54

Brunet, L. R., Carroll, K., Summerton, L., Wagle, S., & Ducreux, M. (2017). Baseline carcinoembryonic antigen predicts response to IMM-101 in advanced pancreatic cancer: Data analysis from a randomised, open label, phase II study. Annals of Oncology28, iii70-iii71.

Dalgleish, A.G., & Dalgleish, A.G. (2022). The role of immune modulation and anti‑inflammatory agents in the management of prostate cancer: A case report of six patients. Oncology Letters, 24, 247. https://doi.org/10.3892/ol.2022.13367

Reply to comment 5 of the second reviewer: Dear Mr/Mrs, thank you for the suggested references. These were reviewed and implemented in the introduction or discussion section of the manuscript when possible. Hereby, the manuscript was improved, thank you.